# Liposomal Doxorubicin, but Not Platinum-Taxane, Supports MHC-II Expression and Immune Maturation in the Ovarian Tumor Microenvironment

**DOI:** 10.3390/cancers17172827

**Published:** 2025-08-29

**Authors:** Hyojae Lee, Xiao-Lei Chen, Duygu Ozmadenci, Elise Tahon, Terrance J. Haanan, Breana Hill, Safir Ullah Khan, Antonia Boyer, David D. Schlaepfer, Dwayne Stupack

**Affiliations:** 1Department of Biology, New York University, New York, NY 10012, USA; 2Moores Cancer Center, University of California San Diego, San Diego, CA 92037, USAdschlaepfer@health.ucsd.edu (D.D.S.); 3Centre de Recherche, Université Laval, Québec City, QC G1V 0A6, Canada

**Keywords:** ovarian tumor, chemotherapy, pegylated liposomal doxorubicin, major histocompatibility antigen

## Abstract

Chemotherapy is now appreciated for its ability to promote immunogenic cell death, yet many studies have not replicated effective preclinical dosing that provides effective tumor control. We used efficacious doses for tumor control and six cycles of a first-line therapy (platinum and taxane, P/T) or a second-line therapy (pegylated liposomal doxorubicin, PLD) and evaluated their impact on the immune and tumor compartments of an aggressive murine model of ovarian carcinoma. Using both flow cytometry and single-cell sequencing, we found that P/T offered excellent tumor control, but limited immune activation. In addition to providing tumor control, PLD improved activation of the immune system, in tandem with upregulation of MHC Class II antigens on the surface of immune cells, ovarian tumor cell lines, and patient-derived tumor cells.

## 1. Introduction

Cytotoxic chemotherapy is complemented in some types of cancer by immunotherapy, particularly by the use of checkpoint inhibitor therapies [1,2,3,4]. Support for the concept of combining immunotherapy and chemotherapy is based on studies that support the concepts that chemotherapy elicits interferon responses [5,6], or leads to tumor death or other altered biologies that result in increased immunogenicity [6,7,8]. As some tumors are recalcitrant to immunotherapy despite displaying immune response markers, a better understanding of the interactions between chemotherapy and the immune system would be valuable.

In the case of ovarian carcinoma, and particularly the high-grade serous subtype, transcriptomic data obtained from patients’ tumors reveal immune markers that are prognostic for outcome [9,10,11,12]. Immunohistochemical assessments similarly support the notion, as patients with infiltration of T cells into the tumor compartment show improved survival [13,14]. Yet despite these promising indications, immunotherapy has been largely unsuccessful in ovarian cancer [15]. This is unfortunate, since standard-of-care chemotherapy also fails most patients. About one-fifth of patients are initially resistant to chemotherapy. Among those that do respond initially, progressive disease occurs within twenty months in about half of the patients. While newer drugs such as PARP inhibitors can push this back, they do not extend the median overall survival [16].

One important caveat when considering the accumulating evidence for chemotherapy boosting immunotherapy, however, is the relatively small fraction of data with clinically relevant chemotherapy dose, combination, and cycles of treatment [17,18,19]. These are not trivial considerations, since cancer patients are treated with maximum tolerable dosing of chemotherapy as a matter of course. Most patients are given combinations of chemotherapies and will ultimately receive six or more cycles of treatment. This is an important consideration because cyclic dosing of therapy is associated with lymphotoxicity, with profound reductions (95%) across several immune populations, including lymphocytes; CD4 T cells and B cells often remain depleted even after 9 months [20]. Paradoxically, concomitant dosing of anti-PD1 with chemotherapy is associated with poorer CD8 T cell effector responses [21]. This contrasts with the use of chemotherapy as an immune adjuvant, where lower doses and careful timing are often considered [22]. The data collectively suggest that chemotherapy may influence the immune system, though direct comparison of efficacious doses of different chemotherapies is rare.

Here, we have performed an initial comparison of chemotherapy effects upon mice bearing aggressive ovarian cancer tumors (KMF) developed within our laboratory. KMF shares gains and losses with HGSC [23] with efficacious and clinically relevant doses of first-line combination chemotherapy (six cycles of platinum and taxane, P/T) or with a second-line therapy (six treatments with pegylated liposomal doxorubicin, PLD), most typically used in patients who are platinum resistant. PLD has been associated with improved immune markers in prior models of ovarian cancer [24,25]. We followed tumor growth and analyzed cell composition of the tumor compartment using flow cytometry and scRNAseq. Both chemotherapeutic approaches proved capable in the overall control of tumor growth. However, there were differences in the tumor cells surviving each chemotherapy, with PLD-treated tumors expressing significantly greater levels of major histocompatibility antigens (MHC class II). This corresponded to a more robust immune response, with an elevated presence of dendritic, T, and B cells, with more subtle effects on macrophage populations. The data, overall, provide a compelling case that chemotherapies have differing capacities to promote an anti-tumor immune response.

## 2. Materials and Methods

### 2.1. Bulk RNA Transcriptomic Analysis

Quantified raw sequencing data were downloaded from https://cbioportal.org (accessed on 12 December 2024) in the available format (FPKM tables). Each of the 20,090 genes with mapped RNA-seq data was classified into one of six categories for cancers based on the FPKM levels in 17 cancer types, respectively: (1) Not detected: FPKM < 1 in all cancers; (2) Enriched: at least a 5-fold higher FPKM level in one cancer than in all other cancers; (3) Group enriched: a 5-fold higher average FPKM value in a group of 2–7 cancers than in all other cancers; (4) Expressed in all: detected in all 32 cancers with FPKM > 1; (5) Enhanced: at least a 5-fold higher FPKM level in one cancer than the average value of all 17 cancers; and (6) Mixed: the remaining genes detected in 1–16 cancers with FPKM > 1 that did not fit the above categories.

Based on the FPKM value of each gene, we classified the patients into two groups and examined their prognoses. In the analysis, we excluded genes with low expression, i.e., those with a median expression among samples less than FPKM 1. The prognosis of each group of patients was examined by Kaplan-Meier survival estimators, and the survival outcomes of the two groups were compared by log-rank tests. To choose the best FPKM cut-offs for grouping the patients most significantly, all FPKM values from the 20th to 80th percentiles were used to group the patients, significant differences in the survival outcomes of the groups were examined, and the value yielding the lowest log-rank *p* value was selected. Genes with log-rank *p*-values less than 0.001 were defined as prognostic genes. In addition, if the group of patients with high expression of a selected prognostic gene has a higher observed event than expected event, it is an unfavorable prognostic gene; otherwise, it is a favorable prognostic gene.

### 2.2. In Vivo Tumor Growth and Harvest of Single-Cell Suspension of HGSOC

KMF, HGS2 and OVCAR8 cells were cultured as described [23,26]. For tumor models, 5 × 10^6^ KMF cells were seeded i.p. in 8–10 week old C57bl/6 mice (~20 g) to mimic disseminated disease. Tumors were allowed to establish for one week, and mice were subsequently treated with chemotherapy. Control treatment groups (10 mice, no therapy) were compared to mice treated with cisplatin (2 mg/kg) and taxane (10 mg/kg) (10 mice) or to PEGylated doxorubicin liposomes (Doxil) (0.15 mg/kg) (10 mice) twice weekly for three weeks (six cycles). Tumor cells and immune cells were harvested from the ascitic tumor microenvironment of six mice for flow cytometry or from two mice for single-cell RNA sequencing. 10× Genomics single cell 3′ RNA Cell barcoding, RT, cDNA amplification and library construction were all performed using Chromium Next GEM Single Cell 3′ HT v3.1 (10× Genomics). The preclinical studies described here were approved by the UC San Diego IACUC, approval S05356 (Approved 29 November 2022).

### 2.3. Single-Cell RNA Seq Data Analysis

Libraries were sequenced on Illumina NovaSeq 6000 at 1 billion reads per sample. The demultiplexed fastq files went through unique molecular identifier (UMI) quantification using 10× CellRanger v6.1.2, and the reads were aligned to the mm10 mouse reference genome.

The aligned reads are then imported to R (v.4.1.2) using Seurat v4.3.0 for secondary analysis. Cells were then filtered using the following criteria: 250–10,000 genes per cell, 1000–100,000 UMIs, less than 20% mitochondrial reads, and less than 40% ribosomal reads as quality control measures. Doublets were filtered using Doubletfinder (2.0.3). Regularized negative binomial regression (SCTransform) was used to maximize biological distinctions between cells. Principal component analysis (PCA) was performed on the integrated data, as well as shared nearest-neighbor (SNN) graph construction, and cluster determination using the SNN graph modularity optimization-based clustering algorithm (all using 40 dimensions, with resolution = 0.4) for initial cell clustering, including all cell types. The top expressed genes for each cluster were calculated using the FindAllMarkers function with the default non-parametric Wilcoxon rank sum test. For initial cell type annotation, the ScType method was used with manual adjustment.

### 2.4. Gene Set Enrichment Analysis (GSEA) and Other Pathway Analysis

For fast pre-ranked gene set enrichment analysis (GSEA), fgsea (v1.20.0) was used along with the following gene sets: Hallmark gene sets (h.all.v7.5.1) and curated gene sets (c2.all.v7.5.1). For the more advanced enrichment analysis using gene ontology (GO) pathway and Hallmark gene set analysis, the following R packages were used: escape, GSEABase and SingleCellExperiment, along with UCell, which is based on the Mann-Whitney U statistic. UCell is a robust way to compare gene signature scores at this dataset size [12].

For the analysis of TCGA human ovarian cancer data, primary tumors and recurrent tumors were compared against each other on UCSC Xena. https://xenabrowser.net (accessed on 1 December 2024). For the gene set enrichment analysis, blitzGSEA was performed with 2000 permutations, 5 minimum number of genes, 500 maximum number of genes and 20 anchors. For differential gene expression analysis, the limma-voom method was used, with a *p*-value threshold of 0.05, logFC threshold of 1.5, 500 maximum genes for Enrichr, and gene ontology pathways.

### 2.5. Flow Cytometry

The peritoneal immune cells (2 million cells) were incubated with Fc block (BD Biosciences, 553141) for 15 min. For surface marker staining, the following antibodies were used in flow staining buffer (PBS supplemented with 2% bovine serum albumin) at 4 °C for 30 min: CD45 (BD Biosciences, Franklin Lakes, NJ, USA; #560510), B220 (Biolegend, San Diego, CA, USA; #103255), TCR beta (Thermo Fisher Scientific, Waltham, MA, USA; #47-5961-82), CD4 (Biolegend, San Diego, CA, USA; #100509), CD8 (BD Biosciences, Franklin Lakes, NJ, USA; #552877), MHC Class II (I-A/I-E) (Thermo Fisher Scientific, Waltham, MA, USA; Invitrogen, #50-112-8850), and MHC Class I (H-2Db) (eBioscience, San Diego, CA, USA; #25-5999-80), CD11b (Biolegend, San Diego, CA, USA; #101233), F4/80 (Thermo Fisher Scientific, Waltham, MA, USA; #45-4801-82), CD11c (Biolegend, San Diego, CA, USA; #117334), NK1.1 (BD Biosciences, Franklin Lakes, NJ, USA; #550627). Flow cytometry was performed using a BD LSR Fortessa and Fortessa X-20, and data were analyzed with FlowJo v10 software.

### 2.6. Statistics

Bioinformatics approaches as described above; for flow cytometry, MFI values or percent positive were compared via ANOVA, with a *p* < 0.05 considered significant.

Study approval.

### 2.7. Consent and Sex as a Biological Variable

These studies were focused on ovarian cancer, which affects females only. The OV-TCGA patient dataset is de-identified and publicly available.

### 2.8. Data Availability

All scRNAseq data will be uploaded to the NCBI GEO database upon acceptance. 

## 3. Results

### 3.1. Immune Cell Transcripts Are Prognostic for Outcome in High-Grade Serous Ovarian Carcinoma

Despite being considered an immunologically “cold” tumor, studies have linked immune markers or the presence of tumor-infiltrating lymphocytes with improved overall response in ovarian cancer [9,10,12,13,27,28,29,30,31,32]. We analyzed the levels of several immune transcripts and outcomes in the TCGA_OV database of high-grade serous ovarian cancer, and subsequently confirmed the associations in the HPA ovarian cancer dataset. Several markers were associated with improved outcome, including the chemokine CXCL13 (Table 1), which we previously associated with improved outcome using preclinical modeling [33]. Interestingly, markers associated with poorer outcomes are frequently associated with myeloid cells. These data are consistent with the concept that the adaptive immune response is important in promoting survivorship. However, as tumor sequencing precedes chemotherapy in these cases, it was not possible to make conclusions about the impact of chemotherapy on immune status.

### 3.2. Chemotherapy Controls Tumor Growth in an Aggressive Murine Model

To gauge how chemotherapy impacts the immune compartment, we first evaluated the impact of either the standard first-line (P/T) chemotherapy or a second-line (PLD) chemotherapy, at efficacious doses, on KMF [23,33] tumors growing orthotopically. Treatment of mice with PLD or P/T yielded comparable tumor control, as assessed by the expression of luciferase, while tumors expanded rapidly in untreated mice (Figure 1A,B). To better understand the tumor-immune landscape, the admixture of tumor and immune cells isolated from the peritoneum was analyzed by single-cell RNA sequencing (Figure 1C), revealing both tumor-associated immune populations that included lymphoid cells, macrophages and dendritic cells. Analysis of the isolated cells by flow cytometry confirmed significant CD45+ cells in all groups, with dramatically decreased populations of tumor cells in either (P/T or PLD) chemotherapy-treated population (Figure 1D).

### 3.3. Chemotherapy Type Is Associated with Variation in Immune Components

While mice treated with PLD or P/T displayed a dramatic decrease in the proportion of tumor (CD45-) cells present in ascites, we noted a variety in the immune cells present. Cells isolated from PLD-treated mice revealed an enrichment of dendritic cells, natural killer cells, B cells, and both CD4 and CD8 T cell populations (Figure 2A–F). NK enrichment displayed the greatest variation by mouse, yet remained significant (*p* < 0.01). By contrast, mice treated with P/T displayed a modest increase selectively in the CD4+ T cell population, but not other populations. Macrophages were the most abundant CD45+ population overall, particularly in the tumor-bearing mice. Macrophage trended towards decreased presence with chemotherapy; the results were not significant.

### 3.4. Transcriptomics Reveal Differences Across CD45+ Populations by Chemo Type

To complement and extend the flow cytometry studies, we next evaluated the subtypes of immune cells (CD45+) using single-cell sequencing approaches (Figure 3A). Consistent with the flow cytometry data, we noted statistically significant changes across several different lymphoid populations (Figure 3B). P/T or PLD was associated with a similar increase in the number of B cells (note that the PLD and P/T B cell enrichment are overlapping points in Figure 3B). The scRNAseq data provided additional evidence for increases in T cells and dendritic cells associated with PLD treatment (Figure 3B). P/T or PLD were associated with a decrease in the neutrophil/polymorphonuclear cell population, while P/T alone negatively impacted the incidence of dendritic cells and progenitor cells. There were additional qualitative differences between the lymphoid populations from PLD and P/T, respectively. Higher levels of T cell activation (CD28, CD69), many bearing regulatory markers (*Foxp3*, *Ctla4*), were present in PLD-treated tumors (Figure 3C). Similarly, the B cells present in PLD-treated tumors showed increased transcripts for MHC class II and *Cd274*, whereas B cells from P/T- treated mice expressed higher levels of *Cd19* and *Igmh*, suggesting a less mature phenotype (Figure 3C).

Although the cells identified as macrophage showed similar relative abundance across treatment groups (Figure 2A and Figure 3B), with controls having the most, though all cell populations were captured. We identified 10 clusters of macrophages by single-cell sequencing (Figure 4A). The macrophages were broadly classified as tissue resident macrophage (*Gata6^HI^*, *Fgfr1^+^*, *Alox5^+^*; cluster 0, 5, 6, 7) or bone marrow derived (BMD, all other groups) (Figure 4B), which included a moderate gata6-expressing group of cells (cluster 9) that were enriched in stress antigens (*S100A8,9*). The Ly6c2-enriched clusters, 1, 2 and 4 were suppressive in nature (expressing inflammation-dampening *Cd300lf*), while populations 3 and 8 expressed abundant *Cd74* and associated MHCII antigens. Stress antigen-macrophage in cluster 9 trended upward with P/T treatment but dramatically decreased (>2.7 fold Log_2_) in PLD (Figure 4C, see split UMAP). For macrophage cluster 3 (MHC-expressing and ECM-producing), the proportion was constant across all three groups. Populations 0–2, 4–5, and 6–8 showed similar patterns with chemotherapy; clusters 1, 2, and 4 were depleted while populations 0 and 5–8 were enriched. In each case, the effect of PLD was somewhat greater, and was pronounced for the CXCL13-expressing tumor resident macrophage cluster 7 and MHC-bearing recruited inflammatory macrophage cluster 8.

### 3.5. Chemotherapy Induces Selective Changes in the Abundance of Tumor Cell Clusters

The tumor cell metaclusters (Figure 5A) contained a stromal population and six general populations of tumor cells, identified as follows: C0, enriched for oxidoreductases and reductases; C1, enriched for proteins associated with actin/membrane dynamics; C2 and C3, enriched for major histocompatibility class (MHC) antigen expression, with C3 also enriched for extracellular matrix (ECM) production; C4, selectively elevated ECM expression; and C5, tumor cells undergoing mitosis by gene set enrichment (Figure 5B). The impact of chemotherapy on these tumor cell populations varied. The population of stromal cells, which were largely tumor-associated fibroblasts (C6), and one population of MHC-expressing tumor cells matrix (C3), were increased relative to untreated controls with either PLD or P/T. PLD also significantly increased a second MHC-expressing cell population (C2) while selectively depleting populations enriched for actin and membrane dynamics (C1). This population (C1) was unaffected by P/T, which instead was more potent at depleting the proliferating tumor cell population (C5). The most significant differences between the chemotherapies used here appeared to be the most sensitive tumor populations and the ability to increase tumor cell expression of MHC antigens.

### 3.6. PLD Selectively Promotes MHCII Expression in Ovarian Cancer Cells

Given the importance of MHC in fostering an immune response, we examined this further. We performed composite scoring of several class I and class II MHC markers. In this case, PLD-treated tumors expressed significantly elevated levels of MHC class II-related transcripts (*p* < 0.001) and a slight increase in MHC class I expression relative to controls or to P/T (*p* < 0.03) (Figure 6A). Examining the tumor cells isolated from P/T- or PLD-treated mice by flow cytometry, we confirmed an enhanced expression of MHC Class II on the cell surface, both in terms of the population expressing and the relative density per cell as measured by geometric mean fluorescence (Figure 6B). To establish whether the induction of MHC II was a direct or indirect effect of PLD, we next evaluated MHC expression in response to chemotherapy treatment in vitro. KMF tumor cells expressed little MHC class II in culture. This was somewhat increased by docetaxel, but was profoundly increased by PLD (Figure 6C). The observed effect was general and could be readily reproduced in other mouse and human ovarian tumor cell lines (Figure 6D). Lastly, we tested whether PLD could induce the expression of MHC II in human tumor cells following ex vivo isolation from patients being treated with P/T chemotherapy. Again, doxorubicin was universally capable of eliciting increased expression of MHC II. In this case, the effect was muted (Figure 6E), but consistent and significant (*p* < 0.01). Altogether, these data support the notion that PLD results in a net activation of immune function, which may contribute to observed tumor control in the absence of a significant impact on the proliferating tumor cell population.

## 4. Discussion

Platinum, taxane and PLD chemotherapies induce immunogenic cell death (as reviewed in [6]), yet only PLD induces tumor MHC expression [24,34,35]. MHC expression has been associated with improved outcomes in ovarian cancer [36], though we did not consistently see this for individual markers in our analysis of the TCGA_OV or HPA validation datasets. Instead, we noted many immune markers, including MHC, that consistently trended (nonsignificant, *p* = 0.5–0.15) towards significance, but did not reach the significance criteria required for inclusion in Table 1. This was the case for a host of other immunomarkers (e.g., *Ccl17*, *Cspg5*, *Mif*, *Fcrl3*, *Xbp1*, *Cd79A*, Cd112, Cd155). The improved MHC expression and increased populations of lymphoid cells associated with PLD have been noted by others previously [24,34,37,38] but not as part of a true, multi-cycle therapeutic regimen, and certainly not with evaluation of changes in individual cell transcripts across the tumor and immune compartment, as we have documented here.

The reason for the relative increase across the immune cell populations s observed in the PLD populations is not completely evident, as doxorubicin is certainly lymphotoxic [39]. The increase may rather reflect the development of doxorubicin as an encapsulated agent in PLD [40,41]. Typically, doxorubicin is highly toxic to lymphoid cells, inducing apoptosis regardless of cell cycle [41,42]. When encapsulated as PLD, it undergoes selective tumor uptake and is less toxic to other tissues [43]. The tumor cell population most depleted by PLD was characterized by enhanced levels of transcripts encoding dynamic membrane/actin system elements (C1), and it is tempting to speculate that this might indicate the most susceptible target population. Conversely, P/T chemotherapy was somewhat more effective at specifically depleting the population of proliferative tumor cells. However, P/T was not as effective at inducing MHC following several cycles of use relative to PLD, nor with other markers of immune maturation.

PLD-induced the expression of MHC class I and class II was not unique to KMF cells. Consistent with prior reports [24,34,35], PLD induced expression across several human and mouse cell lines, including the BRCA1-deficient HGS2 cells shown here. PLD also increased MHC class II expression in tumors isolated from patients who were undergoing treatment with P/T, though with muted impact. This may be due to the effects of P/T on patients. The observation that P/T treatment was associated with the presence of fewer lymphoid cells than PLD was consistent with broader uptake of the drug and a general effect of P/T upon all cell types. This is not inconsistent with the immunogenic effects of chemotherapy in promoting an anti-tumor response. Rather, the simplest interpretation is that this reflects the balance of chemotherapy-induced immunogenic cell death with lymphotoxicity. While both platinum agents and taxanes do cause immunogenic cell death and do stimulate the immune system, multiple cycles of curative-intent dosing of a combined chemotherapy regimen may nonetheless promote chemotoxic effects within the immune compartment. Alternative explanations could relate to differences in the pharmacokinetic profiles of the drugs. Encapsulated doxorubicin (PLD) offers advantages in exposure relative to free drug, with a 20+ fold increase in AUC, which may provide an advantage, despite the significantly higher dosing of P/T. However, PK effects alone are inconsistent with the selective effect on tumor populations shown (Figure 5), and PLD is known to retain immunotoxic activity [41]. Nonetheless, PLD as dosed appears to be more permissive to lymphoid maturation, consistent with MHC antigen induction.

Beyond the focused scope of this study, additional questions arise. One of these is how these two different chemotherapy regimens might best combine with immunotherapy in the clinic. It is tempting to speculate that immune checkpoint inhibitors, for example, may combine better with PLD than with P/T, based on MHC expression and on the increased ratio of lymphocytes to tumor cells. On the other hand, timing a limited number of cycles of neoadjuvant chemotherapy with a subsequent “chemotherapy holiday” that could be associated with immunotherapy may offer an easier clinical path forward without trying to dramatically alter front-line therapy. A second potential concern that arises from the results is whether treatment with P/T in the first line is actually undermining second-line therapies, particularly in those patients who recur quickly. Given that platinum and taxane chemotherapy in breast cancer can result in sustained suppression of T and B cells [20], it is possible that secondary immunotherapy may lack critical immune components, or reactivity, after first-line failure of P/T.

Lastly, individual patients likely differ in the impact that chemotherapy has on their individual immune compartment, just as they vary in the impact on the tumor. Evidence suggests that patients whose immune system survives chemotherapy better will do better overall [44]. It may be interesting to determine biomarkers across the groups of patients with “good” immune signatures (assumed as a pre-requisite) for outcomes, to determine whether critical ancillary factors might be identified.

## 5. Conclusions

Overall, these studies lend credence to the notion that chemotherapies are not interchangeable. We find that PLD may be superior in sparing the immune compartment and in the activation of MHC antigen expression. Given that immune markers as measures prior to chemotherapy are a key indicator of outcome, this may have important translational importance should we reconsider future standards of care.

## Figures and Tables

**Figure 1 cancers-17-02827-f001:**
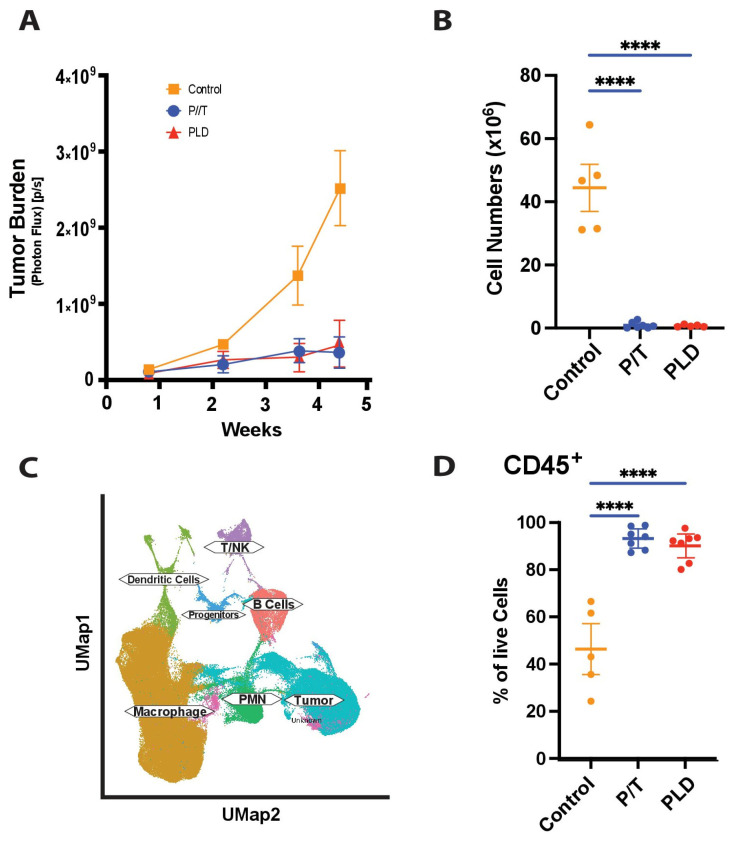
Tumor Growth and Composition. The growth of KMF tumors treated twice weekly with either cisplatin and docetaxel or with pegylated liposomal doxorubicin, or left untreated, was tracked by photon flux from the expressed luciferase gene tracer (**A**) over four weeks. Cell populations isolated from the peritoneal cavity at the termination of the study were quantified (**B**) to complement the luciferase studies. Isolated cells were also evaluated by single-cell RNA sequencing, with UMAP-depicted clustering (**C**) of isolated tumor and immune cells shown. The significant quantities of immune cells suggested by single-cell sequencing were confirmed by flow cytometry (**D**), with cells isolated and stained with anti-CD45 to determine the relative fraction of cells isolated that expressed the immune marker. CD45-negative populations include both tumor and isolated stromal cells, principally fibroblasts (as detailed below). (**** *p* < 0.0001).

**Figure 2 cancers-17-02827-f002:**
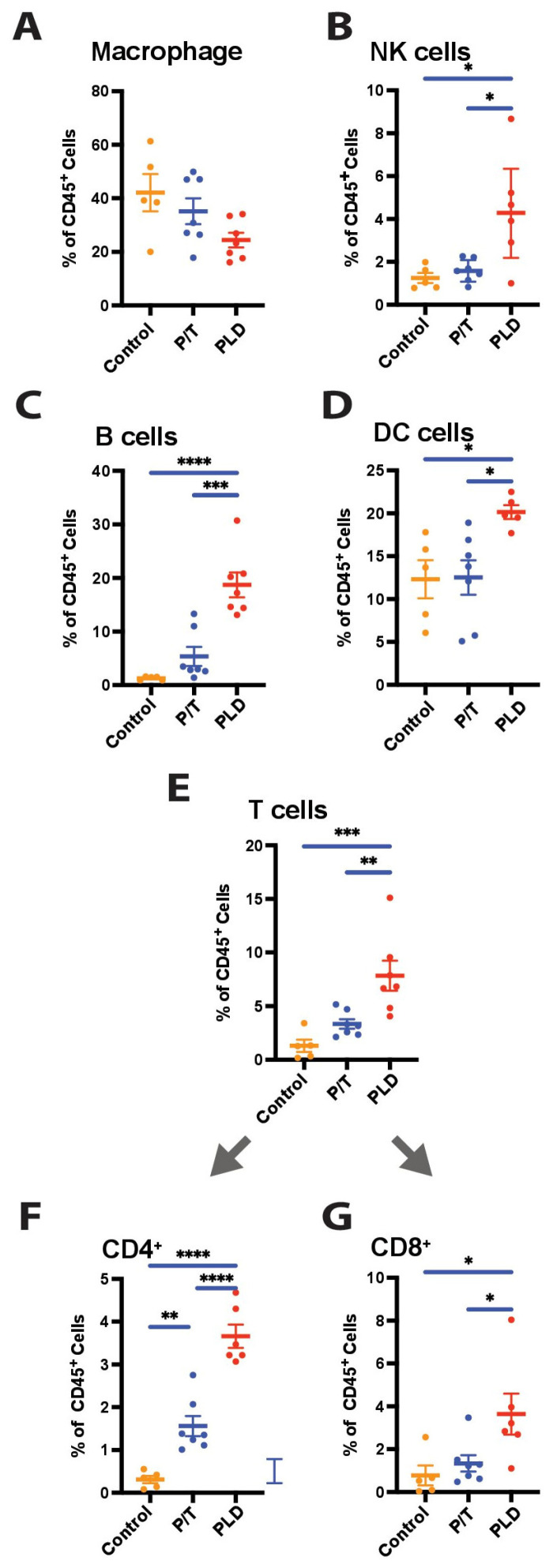
Flow Cytometry Analysis of Isolated Tumor and Immune Populations. Cells isolated from the peritoneum of control mice or mice treated with chemotherapy (P/T or PLD) were incubated with anti-CD45 as well as (**A**) Cd11b (ITGAM) and F4/80 to identify macrophage, (**B**) NK1.1 to identify NK cells, (**C**) anti-B220 to identify B cells, (**D**) CD11C (ITGAX) to identify DC cells, or (**E**) anti-TCR beta to identify T cells, which were also costained with (**F**) anti-CD4 or (**G**) CD8 to identify these subpopulations. (* *p* < 0.05, ** *p* < 0.01, *** *p* < 0.001, **** *p* < 0.0001).

**Figure 3 cancers-17-02827-f003:**
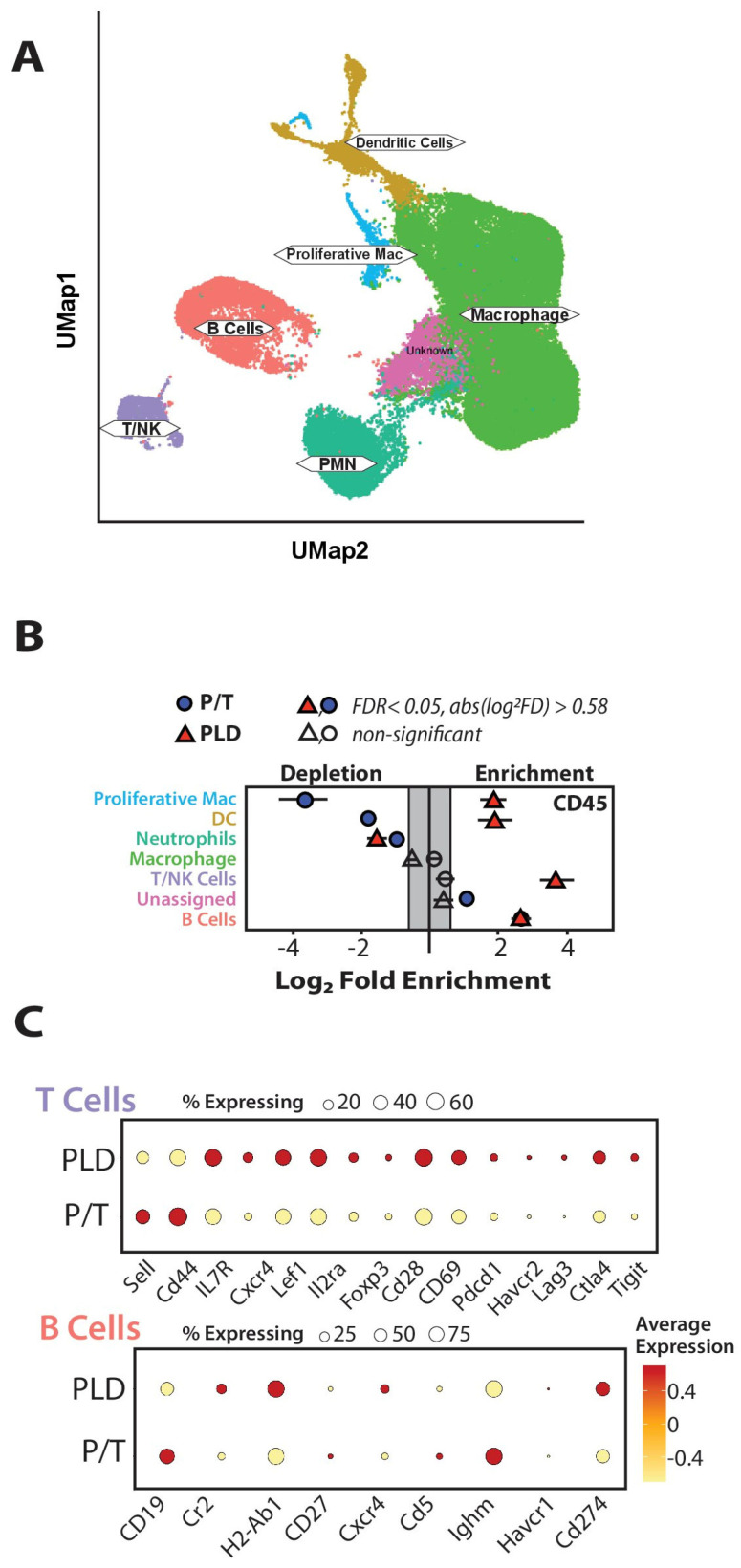
Chemotherapies exert differing pressure on immune populations. The single-cell sequencing-based metacluster of CD45+ cells (**A**) was analyzed to evaluate whether differences occurred in populations. Except for macrophage, each major subtype of immune cells was significantly different from untreated controls (**B**) (False Discovery rate < 0.05, fold difference > 0.58 Log^2^) occurring in each population except macrophage. P/T treated cells are indicated in blue circles, and PLD in red triangles. Note the large increases in T and B cells with PLD and in B cells with P/T. (**C**) Comparison of contrasting individual cellular markers enriched on T and B cell populations based on chemotherapy treatment.

**Figure 4 cancers-17-02827-f004:**
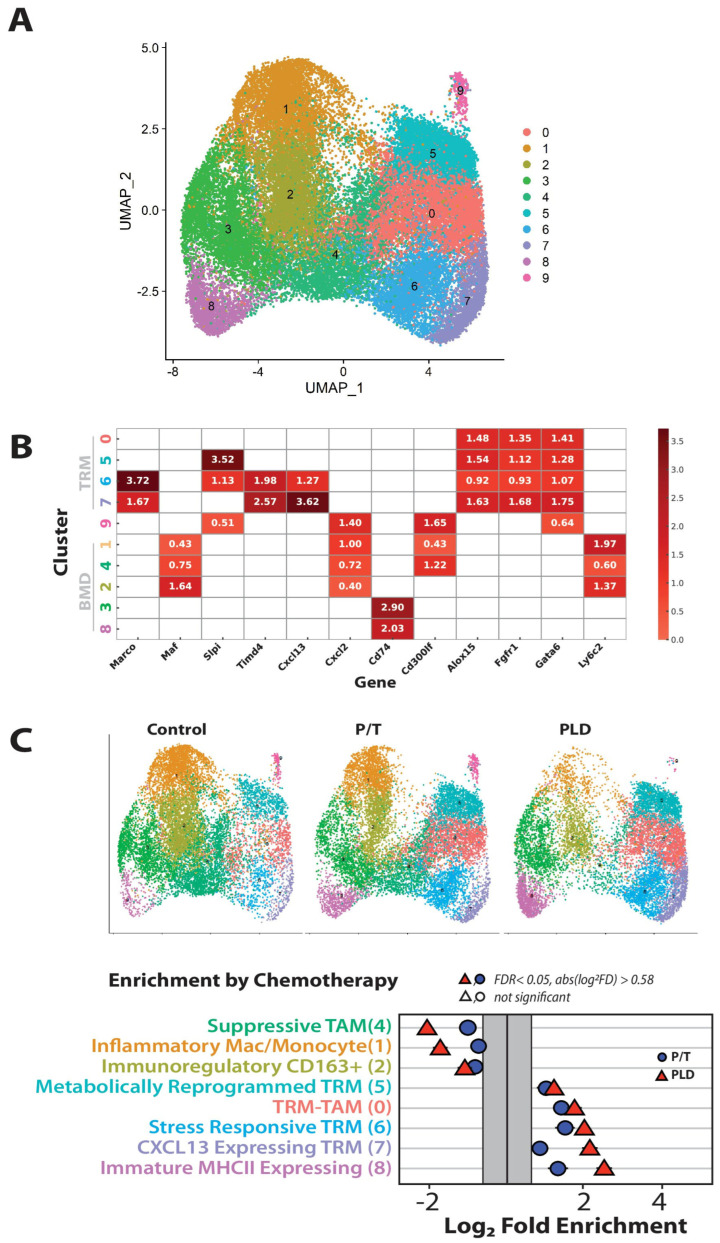
Chemotherapy enriches subtypes of macrophage. Despite a lack of changes in proportional abundance, the subtypes of macrophage present varied, as assessed by single-cell transcriptomic analysis. Macrophage populations clustered into 10 groups (**A**), with the expression of notable genes shown (**B**) in the heatmap dividing macrophage into tissue resident (Gata6+) (TRM) or bone marrow derived (BMD). *Cd74* is a reporter of MHC class II antigen expression. (**C**) The relative enrichment of subpopulations is illustrated in the split umaps. There was a gross change in cluster 9 (stress-activated S100A8/9+) with PLD, and no changes in population 3. All other populations showed significant changes with chemotherapy (False Discovery rate < 0.05, fold change > 0.58 Log^2^).

**Figure 5 cancers-17-02827-f005:**
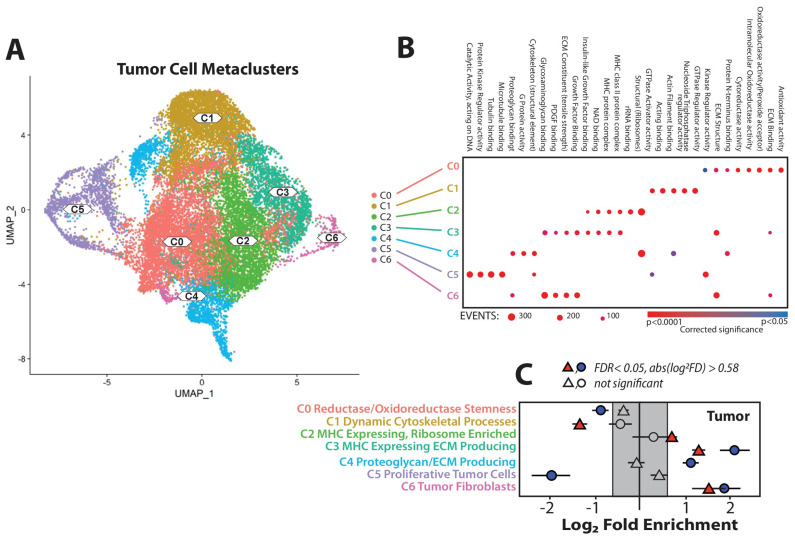
Platinum taxane and liposomal doxorubicin enrich for differing populations of tumor cells. The tumor cell metacluster (**A**) generated from single cell RNAseq depicts seven populations of cells, six tumor (C0–C5) and one fibroblastoid (C6). Significant enrichment or depletion of each cluster, relative to control, by P/T treatment (blue circles), or by PLD (red triangles), is shown (**B**) (False Discovery rate < 0.05, fold difference > 0.58 Log^2^). Each subpopulation varies by chemotherapy type (**C**), though both display an elevated the fibroblast population.

**Figure 6 cancers-17-02827-f006:**
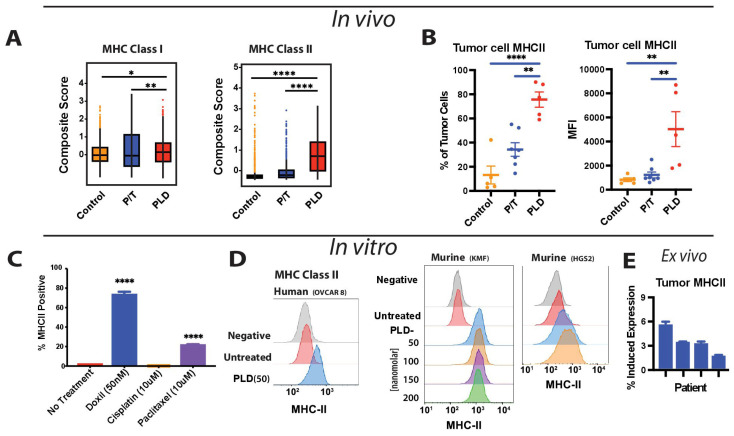
Selective Enrichment of MHC expression by PLD. (**A**) The composite enrichment of MHC genes following platinum and taxane or liposomal pegylated doxorubicin treatment is shown when analyzed by scRNAseq. (**B**) Flow cytometry evaluation of tumor cells isolated from mice treated with P/T or PLD, with the relative fraction of positive cells as well as the mean geometric fluorescence intensity (MFI) shown. (**C**) KMF cells were incubated overnight with chemotherapy agents at concentrations as shown, and cells were subsequently analyzed for the expression of MHC class II by flow cytometry. (**D**) The corresponding histograms for KMF cells treated with doxorubicin, as well as human OVCAR8 or the BRCA1-deficient cell line HGS2, are shown following treatment overnight at concentrations as shown. (**E**) Four isolates of patient-derived tumor cells (all being treated with platinum and taxane chemotherapy) were treated with doxorubicin overnight (40 ng/mL), and the expression of MHC II was assessed by flow cytometry. All patients showed significant increases (*p* < 0.01) relative to controls. (* *p* < 0.05, ** *p* < 0.01, **** *p* < 0.0001).

**Table 1 cancers-17-02827-t001:** Immune Genes Predictive of Outcome.

Gene	Significance		Outcome
	TCGA (n = 349)	Validation (n = 81)	
** *CXCL13* **	0.0023	0.0042	Better
** *CXCL11* **	0.00000025	0.012	Better
** *CXCL9* **	0.014	0.038	Better
** *CD2* **	0.019	0.049	Better
** *CD3* **	0.016	0.04	Better
** *CD6* **	0.0047	0.0047	Better
** *CD9* **	0.026	0.026	Better
** *CD274* **	0.0089	0.011	Better
** *CCL25* **	0.05	0.009	Better
** *CXCR6* **	0.0039	0.021	Better
** *SLAMF7* **	0.00002	0.040	Better
** *BATF2* **	0.001	0.02	Better
** *ISG20* **	0.0001	0.03	Better
** *GZMB* **	0.006	0.0006	Better
** *LAG3* **	0.046	0.0003	Better
** *TIGIT* **	0.0064	0.15	Better
** *NFKBIA* **	0.05	0.001	Better
** *ZAP70* **	0.041	0.13	Better
** *CXCL10* **	0.0008	0.07 *	Better
** *PLAG2D* **	0.008	0.06 *	Better
** *BTLA* **	0.064 *	0.034	Better
** *BTN3A3* **	0.0023	0.082 *	Better
** *CD8A* **	0.046	0.084 *	Better
** *CXCL12* **	0.0044	0.011	Poorer
** *CX3CR1* **	0.003	0.005	Poorer
** *CLEC5A* **	0.00028	0.012	Poorer
***PDGFRA* ****	0.012	0.0037	Poorer
** *BANK1* **	0.033	0.066 *	Poorer
** *PI3* **	0.00002	0.08 *	Poorer

Immune genes detected in bulk RNAseq data that are independently prognostic of outcome across two independent ovarian cancer datasets (TCGA_OV, HPA OV Validation) at *p* < 0.05 or for samples labelled with an asterisk (*) *p* < 0.1 in one of the datasets, as indicated. ** Primarily a Fibroblast/Fibrocyte marker.

## Data Availability

Tcga_ovca dataset is available via https://cbioportal.org (tcga_ovca dataset), and the validation dataset was accessed at https://proteinatlas.org. Single Cell data set upload to NCBI is pending.

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
