# Peer review of "Liposomal Doxorubicin, but Not Platinum-Taxane, Supports MHC-II Expression and Immune Maturation in the Ovarian Tumor Microenvironment"

_cancers, 2025, doi:10.3390/cancers17172827_

Round 1

Reviewer 1 Report

Comments and Suggestions for Authors

This paper investigates whether pegylated liposomal doxorubicin (PLD) promotes MHC II expression and immune cell maturation more effectively than platinum–taxane (P/T) in the ovarian cancer tumor microenvironment. Using a murine ovarian cancer model, the authors administered chemotherapy, harvested tumors and immune cells, and performed single-cell RNA sequencing (scRNAseq) and flow cytometry to assess cellular composition and gene expression changes. These findings were further validated through in vitro cell line experiments and ex vivo analysis of patient-derived tumor cells. So, it is suitable for publication in the journal "Cancers" since it has the interesting topic and results. However, it has the following revised parts. They should be checked prior to the publication. Followings are recommended for the revision. 

Major Revision

  1. In Figures 2–4, the differences in immune cell changes between the PLD and P/T groups could potentially be attributed to disparities in drug exposure rather than inherent pharmacodynamic effects. Inclusion of pharmacokinetic (PK) data—specifically, a comparison of immune cell exposure (AUC) between groups—is necessary to support the interpretation.
  2. The Discussion and Conclusion should be presented as separate sections in accordance with the Cancers journal formatting guidelines.

Minor revison

  1. Page 2, Line 70: Correct the typographical error from concomittant to concomitant.
  2. Page 2, Line 88: Revise “have differing capacity” to “have differing capacities” to agree with the plural subject Chemotherapies.
  3. Page 3, Line 109: Change “P value is selected” to “P value was selected” for past tense consistency.
  4. Page 3, Line 114: Remove the duplicated “of”

Author Response

The reviewer found that the manuscript was “an interesting topic and results” and we are grateful for the reviewer’s appreciating the work.  As the reviewer understands, the manuscript represents a significant amount of work.  The reviewer had several suggestions to improve the paper, which we address below.

  1. In Figures 2–4, the differences in immune cell changes between the PLD and P/T groups could potentially be attributed to disparities in drug exposure… rather than inherent pharmacodynamic effects. Inclusion of pharmacokinetic (PK) data—specifically, a comparison of immune cell exposure (AUC) between groups—is necessary to support the interpretation.

This is a great point and a nice indicator of reviewer expertise. In fact we discussed this point while we were actually writing the manuscript.  How much should we emphasize the encapsulation, and subsequent increase in AUC (~20 fold relative to free dox) as a potential mechanism of action?  Ultimately we did not emphasize this for several reasons: First, the dosing was standardized based on effect, with both treatments controlling tumors similarly, despite the large disparities in drug administration. Second, it was meant to mimic was being done clinically, so ultimately the translation of this work was dependent upon clinical correlation, regardless of actual AUC.  Thirdly, we did not observe stepwise changes in tumor subpopulation effects – ie., hey were distinct effects on different populations of cells (which indicate that while there may be differences in PK, there are also difference in PD).  Thus, an experiment that would measure the exposure of immune populations to the drug is somewhat outside the focus of these studie.  Such a study, if done right, would be a manuscript in and of itself!

Nevertheless, we appreciate the concern.  To address this, we have now added the following paragraph that discusses the potential exposure of cells to the chemotherapy provided,

Alternative explanations could relate to difference in the pharmacokinetic profiles of the drugs.  Encapsulated doxorubicin (PLD) offers advantages in exposure relative to free drug, with a 20+ fold increase in AUC, that may provide an advantage, despite the significantly higher dosing of P/T.  However, PK effects alone are inconsistent with the selective effect on tumor populations shown (Figure 5), and PLD is known to retain immunotoxic activity (41). Nonetheless, PLD as dosed appears to be more permissive to lymphoid maturation consistent with MHC antigen induction. 

The Discussion and Conclusion should be presented as separate sections in accordance with the Cancers journal formatting guidelines.

We thank the reviewer for point this out.  We have now altered the sections so that the conclusion is not set apart beneath its own subheading.

The reviewer has also pointed out the following minor revisions, which have now all been addressed exactly as suggested.  

  1. Page 2, Line 70: Correct the typographical error from concomittant to concomitant.
  2. Page 2, Line 88: Revise “have differing capacity” to “have differing capacities” to agree with the plural subject Chemotherapies.
  3. Page 3, Line 109: Change “P value is selected” to “P value was selected” for past tense consistency.

Reviewer 2 Report

Comments and Suggestions for Authors

Hyojae Lee et al. investigated the immunomodulatory effects of cisplatin, docetaxel, and pegylated liposomal doxorubicin (PLD) in ovarian cancer. Their findings show that PLD but not cisplatin- taxane enhances MHC-II expression and promotes immune maturation within the ovarian cancer microenvironment. Immune cell enrichment was evaluated using multiple approaches, including RNA transcriptomic analysis, in vivo models, single-cell RNA sequencing, and flow cytometry. While the study offers supportive evidence for the field, its novelty is limited since liposomal doxorubicin is already known to upregulate MHC and Fas. Nevertheless, the work reinforces the immunological relevance of PLD.

Comments:

  • Relocate section 2.7 Mouse studies (Ethical statement) before the mouse tumor model.
  • Specify mouse strain, age, and weight at the time of tumor induction.
  • Provide sample size details for in vivo studies.
  • Consider revising the title for clarity and precision.  Liposomal doxorubicin, but not platinum-taxane, supports MHC-II expression and immune maturation in the ovarian cancer tumor microenvironment (ovarian cancer microenvironment or ovarian tumor microenvironment)

Author Response

Response to Reviewer 2

Reviewer 2 noted that, from a novelty perspective, doxorubicin is already known to upregulate MHC, but appreciated the multiple approaches used, the supportive evidence generated, and the impact the work could have on the immunological relevance of PLD.  The reviewer had the following suggestions to improve the manuscript:

Relocate section 2.7 Mouse studies (Ethical statement) before the mouse tumor model.

We have now relocated this as suggested.

Specify mouse strain, age, and weight at the time of tumor induction.

We now more clearly state that the mice are 8-10 week (20g) when the tumors are seeded, consistent with our prior studies using this model.

Provide sample size details for in vivo studies.

For the in vivo studies, each group contained ten mice. At harvest, six from each were processed for flow cytometry and two were processed for single cell seq analysis as indicated. The re-written section is reproduced here:

For tumor models, 5x106 KMF cells were seeded i.p. in 8-10 week old C57bl/6 mice (~20g) to mimic disseminated disease. Tumors were allowed to establish for one week, and mice were subsequently treated with chemotherapy. Control treatment groups (10 mice, no therapy) were compared to mice treated with cisplatin (2mg/kg) and taxane (10 mg/kg)(10 mice) or to PEGylated doxorubicin liposomes (Doxil) (0.15mg/kg)(10 mice) twice weekly for three weeks (six cycles).  Tumor cells and immune cells were harvested from the ascitic tumor microenvironment of six mice for flow cytometry or from two mice for single cell RNA sequencing. 10x Genomics single cell 3’ RNA Cell barcoding, RT, cDNA amplification and library construction were all done using Chromium Next GEM Single Cell 3ʹ HT v3.1 (10x Genomics).  The preclinical studies described here were approved by the UC San Diego IACUC, approval S05356. 

Consider revising the title for clarity and precision. 

We thank the reviewer. We will use the second suggested title: "Liposomal doxorubicin, but not platinum-taxane, supports MHC-II expression and immune maturation in the ovarian tumor microenvironment."